# Diagnostic Value of ^18^F-FDG PET/MRI for Revised 2018 FIGO Staging in Patients with Cervical Cancer

**DOI:** 10.3390/diagnostics11020202

**Published:** 2021-01-29

**Authors:** Hideaki Tsuyoshi, Tetsuya Tsujikawa, Shizuka Yamada, Hidehiko Okazawa, Yoshio Yoshida

**Affiliations:** 1Department of Obstetrics and Gynecology, University of Fukui, Fukui 910-1193, Japan; sikeno@u-fukui.ac.jp (S.Y.); yyoshida@u-fukui.ac.jp (Y.Y.); 2Biomedical Imaging Research Center, University of Fukui, Fukui 910-1193, Japan; awaji@u-fukui.ac.jp (T.T.); okazawa@u-fukui.ac.jp (H.O.)

**Keywords:** ^18^F-FDG PET/MRI, CT, MRI, cervical cancer, revised 2018 FIGO staging

## Abstract

Purpose: To evaluate the diagnostic potential of PET/MRI with ^18^F-fluorodeoxyglucose (^18^F-FDG) in cervical cancer based on the revised 2018 International Federation of Gynecology and Obstetrics (FIGO) staging system. Materials and Methods: Seventy-two patients with biopsy-proven primary cervical cancer underwent pretreatment ^18^F-FDG PET/MRI, CT, and pelvic MRI. The diagnostic performance of ^18^F-FDG PET/MRI and MRI for assessing extent of the primary tumor and ^18^F-FDG PET/MRI and CT for assessing nodal and distant metastases was evaluated by two experienced readers. Histopathological and follow-up imaging results were used as the gold standard. McNemar test was employed for statistical analysis. Results: Accuracy for the invasion of vagina, parametrium, side wall, and adjacent organs was 97.2%, 93.1%, 97.2%, and 100% for ^18^F-FDG PET/MRI; and 97.2%, 91.7%, 97.2%, and 100% for pelvic MRI, respectively (*p* > 0.05). Patient-based accuracy for metastasis to pelvic and paraaortic lymph nodes and distant organs was 95.8%, 98.6%, and 100% for ^18^F-FDG PET/MRI; and 83.3%, 95.8%, and 97.2% for CT, respectively; metastasis to pelvic lymph nodes was statistically significant (*p* < 0.01). Lesion-based sensitivity, specificity, and accuracy for lymph nodes were 83.3%, 95.9%, and 94.8% for ^18^F-FDG PET/MRI; and 29.2%, 98.9% and 93.1% for CT, respectively; sensitivity was statistically significant (*p* < 0.001). After excluding patients diagnosed by conization, accuracy for revised FIGO staging 2018 was significantly better for ^18^F-FDG PET/MRI (82.1%) than for CT and MRI (60.7%) (*p* < 0.01). Conclusions: ^18^F-FDG PET/MRI offers higher diagnostic value for revised 2018 FIGO staging, suggesting that ^18^F-FDG PET/MRI might provide an optimal diagnostic strategy for preoperative staging.

## 1. Introduction

Cervical cancer is the fourth most common cancer in women worldwide, with more than half a million cases diagnosed annually, even though prophylactic vaccines against human papillomavirus have reduced the occurrence of this disease [1]. Previously, cervical cancer had been clinically staged based on the 2009 International Federation of Gynecology and Obstetrics (FIGO) classifications [2,3]. Although preoperative assessment of disease is essential, the 2009 FIGO system did not yet include descriptions of imaging findings other than hydronephrosis and distant metastasis. Moreover, no information was included regarding lymph node involvement, which is associated with poor prognosis. The 2018 FIGO system highlighted the utility of imaging and permitted its use, when available, as part of clinical staging [4,5]. Compared with the 2009 classifications, the revised system better recognizes metastatic or recurrence risk by including a greater number of tumor size subdivisions (IB1, IB2, and IB3) and by taking into account the status of regional lymph nodes detected radiographically or pathologically (IIIC1 and IIIC2). Therefore, the revised system has afforded a greater importance to imaging findings in the planning of optimal treatment.

Cervical cytology and biopsy are the standard methods internationally for detecting and diagnosing disease. Imaging is less useful in screening and for evaluation of microinvasion, particularly in early-stage cancers, such as FIGO IA and part of IB1, which is detectable only by microscopy. When we assess local disease by including tumor size (IB1, IB2, and IB3), vaginal and parametrial invasion (IIA, IIB, IIIA, and IIIB), and the exclusion of bladder and rectal invasion (IVA), MRI provides higher sensitivity and comparable specificity compared with clinical assessment and could be one of the most reliable imaging modalities for assessing local extension in cervical cancer [6,7,8,9].

Lymph node metastasis was not included in the former FIGO classifications, although it is strongly associated with poor prognosis and 10–30% of patients have lymph node metastases on histology even in early cervical cancer [10]. In the revised 2018 FIGO staging system, the presence of regional lymph node metastases is designated as stage IIIC, suggesting the importance of accurate detection by the most suitable imaging modality. CT has been widely used for this purpose because of its cost- and time-effectiveness, whereas ^18^F-fluorodeoxyglucose (^18^F-FDG) PET/CT and MRI each have greater sensitivity and specificity than CT for detecting node metastasis. Moreover, the sensitivity and specificity of ^18^F-FDG PET/CT for aortic node metastasis are greater than those for MRI and CT, suggesting that ^18^F-FDG PET/CT is a valid alternative for detection of lymph node metastasis in cervical cancer [11,12].

The new PET modality of ^18^F-FDG PET/MRI provides high soft-tissue contrast along with the functional imaging of FDG uptake, and has shown potentially better diagnostic performance compared with conventional imaging. Integrated PET/MRI provides comparable diagnostic value in evaluating local extent and better sensitivity and specificity in the detection of nodal metastasis compared with MRI in cervical cancer [13,14,15], which suggests that integrated PET/MRI may have a critical role to play in preoperative diagnosis based on the revised 2018 FIGO staging system for cervical cancer.

The aim of our study is, thus, to evaluate the diagnostic utility of integrated ^18^F-FDG PET/MRI for the revised 2018 FIGO staging in cervical cancer, and to compare the diagnostic accuracy of integrated ^18^F-FDG PET/MRI with those of CT and MRI.

## 2. Materials and Methods

### 2.1. Patients

We retrospectively reviewed the medical records of 72 patients (mean age, 53.0 years; age range, 29–91 years) with biopsy-proven primary cervical cancer. They had undergone ^18^F-FDG PET/MRI, CT, and pelvic MRI with obtained informed consent for the initial staging based on the Japanese Imaging Guidelines of the Japan Radiological Society between November 2015 and June 2020. Patients had completed ^18^F-FDG PET/MRI, CT, and MRI within 3 months prior to treatment. The maximum interval among ^18^F-FDG PET/MRI, CT, and MRI was 109 days (mean, 7.6 days; range, 0–109 days). Two patients underwent conization only, five patients underwent simple hysterectomy, and 39 underwent radical hysterectomy; 39 underwent pelvic and three underwent pelvic and para-aortic lymphadenectomy. Six patients underwent neoadjuvant chemotherapy, 18 underwent definitive radiotherapy with or without chemotherapy, and two refused definitive treatments. This was a multi-center study, as 27 patients with data from CT and/or MRI were referred from other institutions. All patients underwent ^18^F-FDG PET/MRI at our institution.

### 2.2. ^18^F-FDG PET/MRI

#### 2.2.1. Whole-Body PET/MRI

Patients fasted for at least 4 h prior to intravenous injection of 200 MBq of ^18^F-FDG. Fifty minutes after injection, patients were transferred to a whole-body 3.0-T PET/MR scanner (Signa PET/MR; GE Healthcare, Waukesha, WI, USA). Anatomical coverage was from the vertex to the mid-thigh. PET acquisition was performed in three-dimensional (3D) mode with 5.5 min/bed position (89 slices/bed) in 5–6 beds with a 24-slice overlap. A two-point Dixon 3D volumetric interpolated T1-weighted fast spoiled gradient echo sequence was acquired at each table position and was used to generate MR attenuation correction (MR-AC) maps. Dixon-based MR-AC classifies body tissues into soft tissue, fat, and air. PET data were reconstructed by ordered subset expectation maximization (OSEM), selecting 14 subsets and three iterations, and post-smoothing with a 3-mm Gaussian filter. Reconstructed images were then converted to semiquantitative images corrected by the injected dose and the bodyweight of the subject as the standardized uptake value (SUV).

#### 2.2.2. Pelvic PET/MRI

After whole-body scanning and a brief break for urination, the patient was repositioned in the PET/MR scanner. The pelvic PET scan was performed as a 3D acquisition in list mode with 15 min/bed position (89 slices/bed) in 1–2 beds with a 24-slice overlap. Regional PET data were reconstructed with OSEM selecting 16 subsets and four iterations, and post-smoothing with a 4-mm Gaussian filter. The reconstructed images were then converted to SUV images. For pelvic MRI, T2-weighted images were acquired in the sagittal, transaxial, and coronal planes, using the following T2-weighted image parameters: TR, 4000–7000 ms; TE, 146 ms; section thickness, 4 mm; section overlap, 0 mm; flip angle, 100°; FOV, 240 × 240 mm; matrix, 384 × 384; two excitations; and bandwidth, 83.3 kHz.

### 2.3. MRI

Pelvic MRI was performed using a 3-T clinical scanner (Discovery MR750; GE Healthcare, Waukesha, WI, USA) in 49 patients. To delineate the anatomy of the pelvis, T2-weighted imaging was performed in the sagittal, transaxial, and coronal planes. The following T2-weighted image parameters were used: TR, 3200–6000 ms; TE, 60–85 ms; section thickness, 4 mm; interval, 1 mm; flip angle, 111°; FOV, 240 × 240 mm; matrix, 320 × 224; two excitations; echo train length, 10; and bandwidth, 62.5 kHz. In 23 patients, MRI was performed at other institutes using a 1.5-T clinical scanner (Magnetom Aera; Siemens Healthineers, or Signa HDe; GE Healthcare).

### 2.4. CT

CT examinations covering the chest, abdomen, and pelvis were performed using a 64-slice multidetector CT scanner (Discovery CT 750HD; GE Medical Systems, Milwaukee, WI, USA).

### 2.5. Image Interpretation

Images were analyzed on a dedicated workstation (Advantage Workstation 4.6; GE Healthcare). Two board-certificated radiologists/nuclear medicine physicians, each with double certifications and specializing in gynecological imaging, evaluated the ^18^F-FDG PET/MRI, CT, and MRI images retrospectively and reached consensus decisions. Images were evaluated for the following: (a) presence of the primary tumor; (b) tumor invasion into the vagina (IIA or IIIA); (c) tumor invasion into the parametrium (IIIA); (d) tumor extension to the pelvic wall (IIIB); (e) pelvic lymph node metastasis (IIIC1); (f) para-aortic lymph node metastasis (IIIC2); (g) tumor extension to adjacent organs such as the bladder or rectum (IVA); and (h) distant metastasis (IVB). The diagnostic performance of ^18^F-FDG PET/MRI and MRI for assessing the extent of the primary tumor and that of ^18^F-FDG PET/MRI and CT for assessing nodal and distant metastases was evaluated. Both readers were blinded to the results of other imaging studies, the histopathological findings, and clinical data. Each dataset was reviewed as the consensus decision of the two readers after a minimum interval of three weeks to avoid any decision threshold bias due to reading-order effects. For CT and MRI interpretation, several previous standard criteria related to primary tumor and nodal or distant metastatic staging of cervical cancer were used as the reference criteria [6,11]. Swollen lymph nodes larger than 1 cm in short-axis diameter were graded as malignant. For ^18^F-FDG PET/MRI interpretations, the classification of lymph nodes as cancer-positive was based on the presence of focally appreciable metabolic activity above that of normal muscle; or asymmetric metabolic activity greater than that of normal-appearing lymph nodes at the same level in the contralateral pelvis, in a location corresponding to the lymph node chains on CT or MRI images, with reference to previous reports [13,14]. Tumor invasion of neighboring structures was decided primarily on the basis of the CT or MRI findings, with reference to the ^18^F-FDG PET findings.

### 2.6. Reference Standard

Histopathological results were used as the standard of reference. Because clinical and ethical standards of patient management do not require surgery or sampling of all detected lesions, a modified reference standard was used for lesions without histopathological sampling to take into account all prior and follow-up imaging. A decrease in size and/or SUVmax under therapy or an increase in size and/or SUVmax without therapy was regarded as a sign of malignancy. PET-negative and inconspicuous lesions with constant size were rated as benign.

### 2.7. Statistical Analysis

The McNemar test was used to determine the statistical significance of differences in the accuracy of staging as determined by PET/MRI, CT, and MRI. Statistical analysis was performed using PRISM version 6.0 software (GraphPad, San Diego, CA, USA). Differences at the level of *p* < 0.05 were considered statistically significant.

## 3. Results

### 3.1. Patients

According to the revised FIGO criteria [4,5], the stage was classified as IA1 in seven, IA2 in one, IB1 in 19, IB2 in six, IIA in one, IIB in six, IIIB in three, IIIC1 in 15, IIIC2 in five, IVA in two, and IVB in seven, including the subclavicular, longitudinal, and inguinal lymph nodes, lung, and thoracic spine. The histopathologic types of primary tumors were squamous cell carcinoma (*n* = 53), adenocarcinoma (*n* = 17), serous carcinoma (*n* = 1), and clear cell carcinoma (*n* = 1). Demographic data for the 72 patients are listed in Table 1.

### 3.2. Primary Tumor Detection

PET/MRI and MRI detected 83.3% (60/72) and 79.2 (57/72), respectively, of the primary tumors (no significant difference, *p* = 0.248).

### 3.3. Revised FIGO Staging

The overall accuracy of revised FIGO staging for ^18^F-FDG PET/MRI and MRI/CT was 69.4% (50/72) and 50.0% (36/72), respectively (significant difference, *p* < 0.001). When we excluded patients diagnosed by conization to focus on the revision points, the accuracy of the revised 2018 FIGO staging was also significantly better for ^18^F-FDG PET/MRI (82.1%, 46/56) than for MRI/CT (60.7%, 34/56) (*p* < 0.01). ^18^F-FDG PET/MRI understaged the actual stage in 17 patients (23.6%), whereas MRI/CT resulted in understaging in 31 patients (43.1%). ^18^F-FDG PET/MRI incorrectly classified eight IA and four IB1 tumors as no tumors; two IIB tumors as IB1; one IIIB tumor as IIB; and two IIIC1 tumors as IB2 and IIA2; whereas MRI/CT incorrectly classified eight IA and seven IB1 tumors as not tumors; two IIB tumors as IB1; one IIIB tumor as IIB; nine IIIC1 tumors as IB1, IB2, IIA2, and IIB; two IIIC2 as IIIC1, and two IVB as IIA2 and IIIC1. ^18^F-FDG PET/MRI or MRI/CT overstaged the actual stage in five patients (6.9%). ^18^F-FDG PET/MRI or MRI/CT incorrectly classified one IB1 tumor as IIA1, two IB2 tumors as IIB and IIIC1, one IIA2 tumor as IIB, and one IIIC1 tumor as IIIC2. Sensitivity, specificity, and accuracy for detecting growth into the vagina were 100%, 94.6%, and 97.2%, respectively, for both ^18^F-FDG PET/MRI and MRI (*p* = 1). Sensitivity, specificity, and accuracy for growth into the parametrium were 88.6%, 97.3% and 93.1% for ^18^F-FDG PET/MRI; and 88.6%, 94.6%, and 91.7% for MRI, respectively (*p* = 1). Sensitivity, specificity, and accuracy for growth into the pelvic wall were 83.3%, 100%, and 97.2%, respectively, for both ^18^F-FDG PET/MRI and MRI (*p* = 1). Figure 1 and Figure 2 show representative images for the detection of invasion of the parametrium and pelvic side wall. Sensitivity, specificity, and accuracy for pelvic lymph node metastasis were 92.3%, 97.8%, and 95.8% for ^18^F-FDG PET/MRI; and 57.7%, 97.8%, and 83.3% for CT, respectively (*p* = 0.008 < 0.01). Sensitivity, specificity, and accuracy for paraaortic lymph node metastasis were 100%, 98.4%, and 98.6% for ^18^F-FDG PET/MRI; and 75.0%, 98.4%, and 95.8% for CT, respectively (*p* = 0.480). Figure 3 and Figure 4 show representative images for the detection of pelvic and paraaortic lymph node metastasis. Sensitivity, specificity, and accuracy for growth into adjacent organs such as the bladder or rectum were 100%, 100%, and 100% for both ^18^F-FDG PET/MRI and MRI (*p* = 1). Sensitivity, specificity, and accuracy for distant metastasis were 100%, 100%, and 100% for ^18^F-FDG PET/MRI; and 71.4%, 100%, and 97.2% for CT, respectively (*p* = 0.480) (Table 2). Figure 5 shows representative images for the detection of distant metastasis.

### 3.4. Lesion-Based Nodal Metastasis

Lesion-based sensitivity, specificity, and accuracy for lymph node metastasis were 83.3%, 95.8%, and 94.7% for ^18^F-FDG PET/MRI; and 29.2%, 98.8%, and 92.9% for CT, respectively. ^18^F-FDG PET/MRI showed significantly greater sensitivity than CT (*p* < 0.001), whereas CT showed significantly greater specificity than ^18^F-FDG PET/MRI (*p* < 0.05). There was no statistically significant difference in accuracy between ^18^F-FDG PET/MRI and CT (Table 3).

## 4. Discussion

To the best of our knowledge, this is the first study to investigate the diagnostic value of ^18^F-FDG PET/MRI for cervical cancer based on the revised 2018 FIGO staging system in comparison with the conventional imaging modalities of MRI and CT. ^18^F-PET/MRI offered significantly superior accuracy to MRI and CT for the revised 2018 FIGO staging of cervical cancer. The accuracy of ^18^F-PET/MRI for metastasis to pelvic lymph nodes was significantly superior to that of CT, whereas the accuracy of ^18^F-PET/MRI was equivalent to those of MRI and CT for the detection of local extent and metastasis to distant organs. These findings suggest that ^18^F-FDG PET/MRI might provide a useful alternative to conventional imaging modalities in staging cervical cancer.

Cervical cancer had previously been clinically staged based on the 2009 FIGO staging system, which did not include imaging in the staging. Unlike in other cancers such as those of the uterine endometrium or ovary, in cervical cancer treatment options (e.g., radical hysterectomy, trachelectomy, or radiation with or without chemotherapy) are applied depending on the extent of disease. Therefore, there has been an increasing necessity for new techniques that can indicate the optimal treatment. The revised 2018 FIGO staging system recommended the use of imaging techniques as well as clinical evaluation or pathological measurement to assess tumor size and extent and detect lymph node metastases. Although MRI is recommended for assessment of the size and extent of the primary tumor, the choice of imaging modality for nodal evaluation has not yet been fixed by FIGO [4]. Therefore, it is necessary to establish the optimal method for accurate preoperative staging.

The spread of cervical cancer into adjacent organs such as the vagina, parametrium, side wall, bladder, and rectum can be better appreciated on MRI than on CT. Reported pooled sensitivities and specificities of MRI for assessing all aspects of local extent have ranged between 0.71–0.88 and 0.86–0.95, respectively [11]. In particular, MRI has shown significantly better pooled sensitivity (84%) for the evaluation of parametrial invasion compared with that of clinical examination (40%), although pooled specificities were comparable between clinical examination and MRI [9]. Despite of the paucity of data, in the hands of experienced operators ^18^F-FDG PET/CT and ultrasound also have shown similar pooled sensitivity and specificity for the evaluation of parametrial invasion [11]. In terms of ^18^F-FDG PET/MRI, Grueneisen et al. showed that PET/MRI could detect all 27 primary tumor lesions in the uterine cervix and correctly determined the T-stage in 23/27 (85%) patients [14]. Moreover, Sarabhai et al. showed that PET/MRI correctly determined the T stage in 45/53 (85%) patients, whereas MRI alone correctly identified the tumor stage in 46/53 (87%) patients [13]. In the present study, accuracy for the invasion of the vagina, parametrium, side wall, and adjacent organs was 97.2%, 93.1%, 97.2%, and 100% for ^18^F-FDG PET/MRI; and 97.2%, 91.7%, 97.2%, and 100% for pelvic MRI, respectively. However, there were no significant differences between these modalities (*p* > 0.05), suggesting that PET/MRI has similar diagnostic accuracy to that of conventional imaging modalities such as MRI, ultrasound, and ^18^F-FDG PET/CT.

In terms of the detection of lymph node metastasis, PET and PET/CT each showed the highest pooled sensitivity (82%) and specificity (95%), whereas CT showed 50% and 92%, and MRI showed 56% and 91%, respectively [12]. A recent meta-analysis has reported high specificity but poor sensitivity for the detection of lymph node metastases for CT, MRI, and PET [11]. The pooled sensitivities and specificities were 0.51 and 0.87 for CT; 0.57 and 0.93 for MRI; and 0.57 and 0.95, for PET, respectively. However, this review purposefully avoided incorporating cutting-edge, non-standard methods that include acquisition techniques such as integrated PET/MRI because they need validation and are unlikely to be available in regions where cervical cancer is most prevalent [11]. Grueneisen et al. have reported sensitivity, specificity, and diagnostic accuracy for the detection of node-positive patients of 91%, 94%, and 93%, respectively [14]. Moreover, Sarabhai et al. have shown sensitivity, specificity, and accuracy of PET/MRI of 83%, 90%, and 87%; whereas the values for MRI alone were 71%, 83%, and 77%, respectively [13]. In other cancers, including nasopharyngeal cancer, ^18^F-FDG PET/MRI has been reported as superior compared with ^18^F-FDG PET/CT for detecting lymph node metastasis [16]. In the present study, accuracy for detecting metastases in the pelvic and paraaortic lymph nodes was 95.8% and 98.6% for ^18^F-FDG PET/MRI, and 83.3% and 95.8% for CT, respectively. Metastasis to pelvic lymph nodes showed significant difference (*p* < 0.01), suggesting that PET/MRI might have superior diagnostic accuracy compared with the conventional modalities for detecting lymph node metastasis.

In terms of distant metastasis, ^18^F-FDG PET/CT is reported to have high diagnostic accuracy (94%) for identifying metastatic lesions, superior to that MRI and CT [17,18]. For detecting bone metastases, ^18^F-FDG PET/CT has better sensitivity than CT and better specificity than MRI [19]. In terms of ^18^F-FDG PET/MRI, Grueneisen et al. showed that PET/MRI correctly identified non-regional lymph node metastases, including inguinal lymph nodes, in 100% of patients [14]. Moreover, Sarabhai et al. reported that PET/MRI showed higher values for the detection of distant metastases (i.e., metastatic spread beyond regional lymph nodes) compared with MRI alone (sensitivity, 87% vs. 67%; specificity, 92% vs. 90%; diagnostic accuracy, 91% vs. 83%) [13]. In other cancers including breast and colorectal cancers, ^18^F-FDG PET/MRI has been reported as superior compared with ^18^F-FDG PET/CT for detecting distant metastasis to such as the liver [16]. In the present study, accuracy for detecting distant metastasis including to the lung, thoracic spine, and non-regional lymph nodes (except the paraaortic lymph nodes) was 100% for ^18^F-FDG PET/MRI and 97.2% for CT, although there was no significant difference. A possible reason for the lack of significant difference could be the small number of events in our study. Although further studies with larger sample sizes are warranted, we consider that PET/MRI possibly has superior diagnostic accuracy compared with conventional modalities for detecting distant metastasis.

In the revised 2018 FIGO staging system that permits the use of imaging data, MRI has been reported as the best imaging modality for assessing primary tumor of diameter >10 mm [6]. In the present study, the accuracy for local staging was >90% for ^18^F-FDG PET/MRI and also for MRI alone. In the assessment of lymph node metastasis, however, ^18^F-FDG PET/MRI had greater accuracy than CT, which can evaluate only the size of lymph nodes. Therefore, the accuracy for 2018 FIGO staging was significantly better for ^18^F-FDG PET/MRI (82.1%) than for CT and MRI (60.7%), which suggests that ^18^F-FDG PET/MRI might have additional diagnostic value compared with conventional imaging modalities such as MRI and CT.

This study had several limitations. First, this investigation used a retrospective design, and not all MRI examinations were performed at our institution. However, our readers re-evaluated the images from other hospitals and were blinded to the initial imaging findings. Second, the sample size was relatively small, and further prospective studies are needed. Third, we could not evaluate histopathological correlations with imaging in patients who had not yet undergone lymphadenectomy. We thus performed node-specific comparisons between imaging and histopathology in all other patients.

## 5. Conclusions

^18^F-PET/MRI offered significantly superior accuracy compared with MRI and CT for the revised 2018 FIGO staging of cervical cancer. The accuracy of ^18^F-PET/MRI for detecting metastasis to pelvic lymph nodes was significantly superior to CT, and the accuracy of ^18^F-PET/MRI was equivalent to that of MRI and CT for the detection of local extent and of metastasis to distant organs. These findings suggest that ^18^F-FDG PET/MRI might provide a useful alternative to conventional imaging modalities in cervical cancer.

## Figures and Tables

**Figure 1 diagnostics-11-00202-f001:**
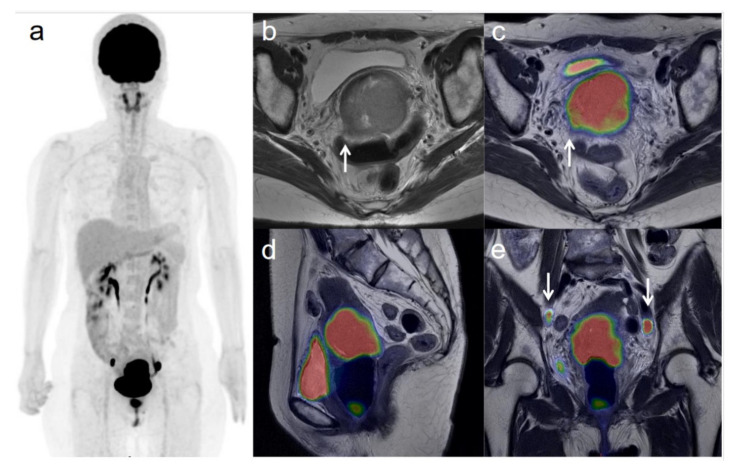
A 54-year-old woman with stage IIIC1 cervical cancer with parametrial invasion. (**a**) ^18^F-FDG PET image shows FDG uptake by tumor in the cervical cervix (arrow) and pelvic lymph nodes. (**b**) Axial T2-weighted pelvic MR image shows parametrial invasion with disruption of the right cervical stroma by the tumor but no extension into the pelvic side wall (arrow). (**c**) Axial T2-weighted PET/MR image shows FDG uptake by the tumor, which invades the right parametrium with disruption of the cervical stroma, but no extension into the pelvic side wall (arrow). These appearances are consistent with the clinical findings. (**d**) Sagittal T2-weighted PET/MR image shows FDG uptake by the cervical tumor. (**e**) Coronal T2-weighted PET/MR image shows FDG uptake by the cervical tumor and pelvic lymph nodes (arrows).

**Figure 2 diagnostics-11-00202-f002:**
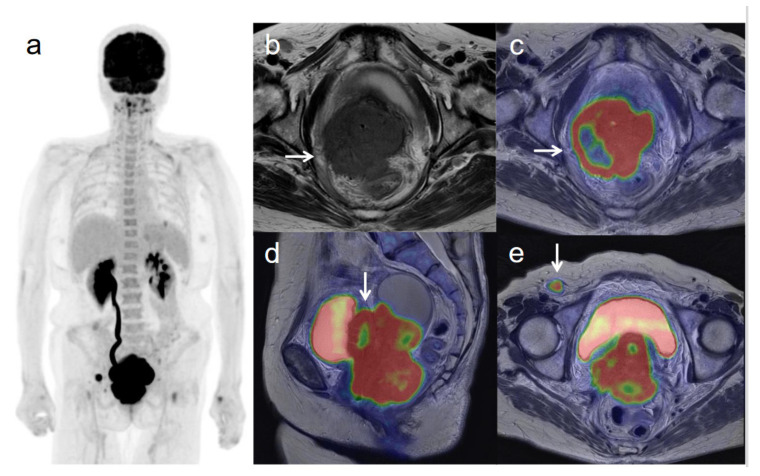
A 62-year-old woman with stage IVB cervical cancer with invasion into the pelvic side wall. (**a**) ^18^F-FDG PET image shows FDG uptake by tumor in the cervical cervix (arrow) and a right inguinal lymph node. (**b**) Axial T2-weighted pelvic MR image shows disruption of the right cervical stroma by the tumor and extension into the pelvic side wall (arrow). (**c**) Axial T2-weighted PET/MR image shows FDG uptake by the tumor, which invades the right cervical stroma and extends into the pelvic side wall (arrow). These appearances are consistent with the clinical findings. (**d**) Sagittal T2-weighted PET/MR image shows FDG uptake by the cervical tumor and invasion into the bladder (arrow). (**e**) Coronal T2-weighted PET/MR image shows invasion of the bladder by the cervical tumor and FDG uptake by a right inguinal lymph node (arrow).

**Figure 3 diagnostics-11-00202-f003:**
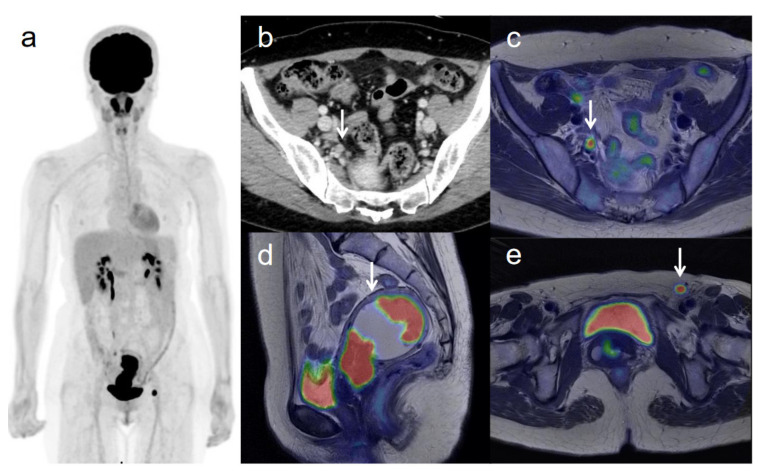
A 53-year-old woman with IVB cervical cancer and pelvic lymph node metastasis. (**a**) ^18^F-FDG PET image shows FDG uptake by tumor in the cervical cervix (arrow) and right pelvic and left inguinal lymph nodes. (**b**) CT shows a right pelvic lymph node of short-axis diameter >1 cm (arrow). (**c**) Axial T2-weighted PET/MR image shows FDG uptake by the right pelvic lymph node (arrow). This finding is strongly suggestive of pelvic lymph node metastasis, which was confirmed by histopathologic examination. (**d**) Sagittal T2-weighted PET/MR image shows FDG uptake by the cervical tumor and invasion into the corpus uteri (arrow), which was confirmed by histopathologic examination. (**e**) Axial T2-weighted PET/MR image shows FDG uptake by the left inguinal lymph node (arrow).

**Figure 4 diagnostics-11-00202-f004:**
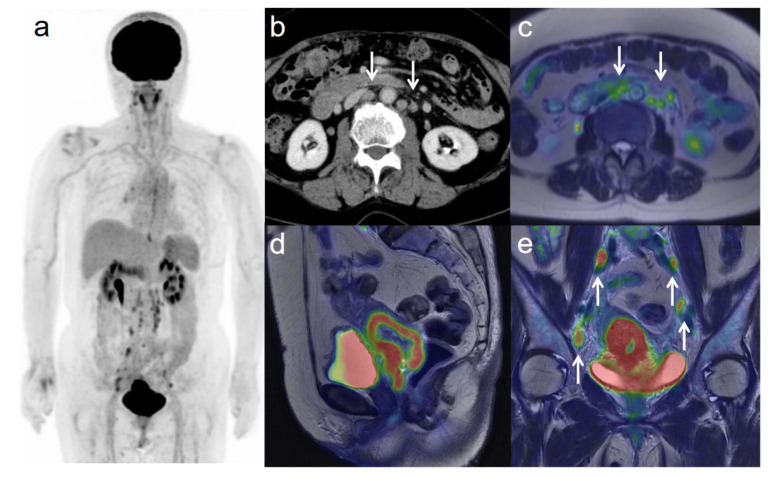
A 75-year-old woman with IIIC2 cervical cancer and paraaortic lymph node metastasis. (**a**) ^18^F-FDG PET image FDG uptake by tumor in the cervical cervix (arrow) and pelvic and paraaortic lymph nodes. (**b**) CT shows paraaortic lymph nodes of short-axis diameter >1 cm (arrows). (**c**) Axial T2-weighted PET/MR image shows FDG uptake by the paraaortic lymph nodes (arrows). This finding is strongly suggestive of paraaortic lymph node metastasis. After radiotherapy, the size and SUV of the lymph nodes decreased, further suggesting malignancy. (**d**) Sagittal T2-weighted PET/MR image shows FDG uptake by the cervical tumor. (**e**) Coronal T2-weighted PET/MR image shows FDG uptake by the cervical tumor and pelvic lymph nodes (arrows).

**Figure 5 diagnostics-11-00202-f005:**
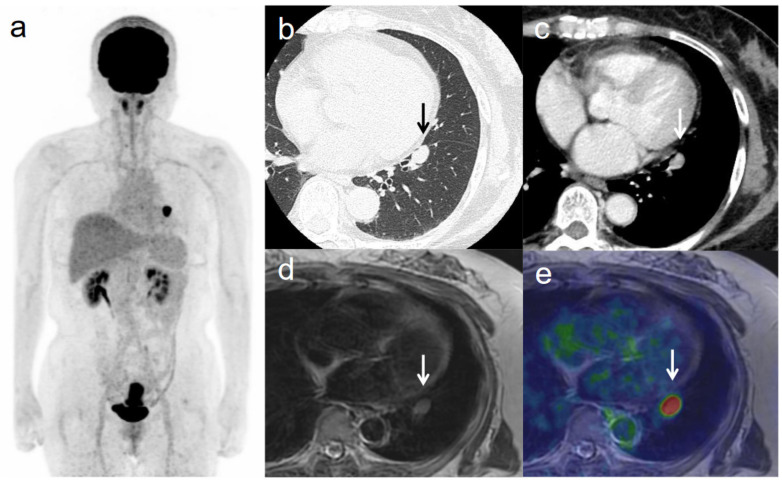
A 63-year-old woman with IVB cervical cancer and lung metastasis. (**a**) ^18^F-FDG PET image shows FDG uptake by tumor in the cervical cervix (arrow) and a lung nodule. (**b**) CT with lung window setting shows a lung nodule of short-axis diameter >1 cm (arrow). (**c**) CT of a lung nodule of short-axis diameter >1 cm (arrow). (**d**) Axial T2-weighted image shows a lung nodule of short-axis diameter >1 cm (arrow). (**e**) Axial T2-weighted PET/MR image shows FDG uptake by the lung nodule (arrow). This finding is strongly suggestive of lung metastasis, which was confirmed by histopathologic examination.

**Table 1 diagnostics-11-00202-t001:** Characteristics of patients with primary uterine cervical cancer.

Case	Age (Years)	Histology	Pathological Staging	2018 FIGO Stage	PET/MRI Staging	MRI and/or CT Staging
1	67	SCC	T3bN1M1	IIIC2	IIIC2	IIIC1
2	45	SCC	T3bN0M0	IIIB	IIIB	IIIB
3	41	SCC	T4aN1M1	IVA	IVA	IVA
4	63	SCC	T2bN0M0	IIB	IIB	IIB
5	37	SCC	T1b2N0M0	IB2	IIIC1	IIIC1
6	65	SCC	T2bN0M0	IIB	IB1	IB1
7	46	AC	T1b1N0M0	IB1	not detected	not detected
8	43	AC	T2bN1M0	IIIC1	IIIC1	IIB
9	31	SCC	T1b2N1M0	IIIC1	IIIC1	IIIC1
10	43	SCC	T1b1N0M0	IB1	IB1	not detected
11	46	SCC	T1a1N0M0	IA1	not detected	not detected
12	42	serous	T2bN1M0	IIIC1	IIIC1	IB1
13	39	SCC	T1b1N0M0	IB1	IB1	not detected
14	45	SCC	T1a1N0M0	IA1	not detected	not detected
15	63	SCC	T2bN0M1	IVB	IVB	IVB
16	59	SCC	T1a1N0M0	IA1	not detected	not detected
17	42	SCC	T1b2N0M0	IB2	IB2	IB2
18	37	AC	T1b1N0M0	IB1	not detected	not detected
19	68	SCC	T3bN0M0	IIIB	IIB	IIB
20	45	SCC	T3bN1M0	IIIC1	IIIC1	IIIC1
21	50	AC	T2bN1M0	IIIC1	IIIC1	IIB
22	43	SCC	T2bN1M0	IIIC1	IIIC1	IIB
23	60	SCC	T4aN1M1	IVB	IVB	IVB
24	70	SCC	T2a2N0M0	IIA2	IIB	IIB
25	77	SCC	T1b1N0M0	IB1	IB1	IB1
26	64	SCC	T1b2N1M0	IIIC1	IB2	IB2
27	68	AC	T3bN1M1	IIIC2	IIIC2	IIIC2
28	68	SCC	T2bN0M0	IIB	IIB	IIB
29	33	SCC	T1b1N0M0	IB1	not detected	not detected
30	79	SCC	T2bN1M1	IVB	IVB	IIIC1
31	35	AC	T2a2N1M0	IIIC1	IIA2	IIA2
32	60	AC	T2bN1M0	IIIC1	IIIC2	IIIC2
33	69	AC	T3bN0M0	IIIB	IIIB	IIIB
34	62	clear	T2bN1M1	IVB	IVB	IVB
35	49	SCC	T4aN0M0	IVA	IVA	IVA
36	65	SCC	T3aN1M1	IIIC2	IIIC2	IIIC2
37	75	SCC	T3bN1M1	IIIC2	IIIC2	IIIC2
38	70	SCC	T2bN0M0	IIB	IIB	IIB
39	32	AC	T1a1N0M0	IA1	not detected	not detected
40	29	SCC	T1a2N0M0	IA2	not detected	not detected
41	51	SCC	T1b1N0M0	IB1	IIA1	IIA1
42	34	AC	T1b2N0M0	IB2	IB2	IB2
43	74	SCC	T2bN1M0	IIIC1	IIIC1	IIB
44	48	SCC	T2bN1M0	IIIC1	IIIC1	IIB
45	32	SCC	T1a1N0M0	IA1	not detected	not detected
46	55	SCC	T1b1N0M0	IB1	IB1	IB1
47	40	SCC	T2bN0M0	IIB	IIB	IIB
48	48	SCC	T1b2N0M0	IB2	IB2	IB2
49	50	SCC	T3bN1M0	IIIC1	IIIC1	IIIC1
50	46	SCC	T1a1N0M0	IA1	not detected	not detected
51	39	AC	T1b1N0M0	IB1	IB1	IB1
52	50	SCC	T1b1N0M0	IB1	IB1	IB1
53	77	SCC	T1b2N0M0	IB2	IIB	IIB
54	91	SCC	T3bN1M1	IIIC2	IIIC2	IIIC1
55	55	SCC	T2bN1M0	IIIC1	IIIC1	IIIC1
56	88	SCC	T1b1N0M0	IB1	IB1	IB1
57	45	SCC	T1a1N0M0	IA1	not detected	not detected
58	75	AC	T3bN0M1	IVB	IVB	IVB
59	56	SCC	T1b1N0M0	IB1	IB1	not detected
60	54	SCC	T2bN1M0	IIIC1	IIIC1	IIIC1
61	66	AC	T2bN0M0	IIB	IB1	IB1
62	38	SCC	T1b1N0M0	IB1	IB1	IB1
63	48	SCC	T1b1N0M0	IB1	IB1	IB1
64	42	AC	T1b1N0M0	IB1	not detected	not detected
65	53	ASC	T2a2N1M1	IVB	IVB	IIA2
66	51	SCC	T1b2N0M0	IB2	IB2	IB2
67	68	SCC	T2bN1M0	IIIC1	IIIC1	IIB
68	31	AC	T1b1N0M0	IB1	IB1	IB1
69	36	AC	T1b1N0M0	IB1	IB1	IB1
70	60	AC	T1b1N0M0	IB1	IB1	IB1
71	30	SCC	T1b1N0M0	IB1	IB1	IB1
72	62	SCC	T4aN1M1	IVB	IVB	IVB

Underline indicates over- or under-diagnosis. SCC, squamous cell carcinoma; AC, adenocarcinoma.

**Table 2 diagnostics-11-00202-t002:** Comparison of ^18^F-FDG PET/MRI with MRI and/or CT for patient-based 2018 FIGO staging and detection of recurrence.

	^18^F-FDG PET/MRI	MRI and/or CT	
Primary tumor			
sensitivity	83.3% (60/72)	79.2% (57/72)	*p* = 0.248
2018 FIGO staging for all patients			
accuracy	69.4% (50/72)	50.0% (36/72)	*p* < 0.001
2018 FIGO staging after excluding patients diagnosed by conization			
accuracy	82.1% (46/56)	60.7% (34/56)	*p* < 0.01
Vaginal invasion (IIA or IIIA)			
sensitivity	100% (35/35)	100% (35/35)	
specificity	94.6% (35/37)	94.6% (35/37)	
accuracy	97.2% (70/72)	97.2% (70/72)	*p* = 1.000
Parametrial invasion (IIB)			
sensitivity	88.6% (31/35)	88.6% (31/35)	
specificity	97.3% (36/37)	94.6% (35/37)	
accuracy	93.1% (67/72)	91.7% (66/72)	*p* = 1.000
Extension to the pelvic wall (IIIB)			
sensitivity	83.3% (10/12)	83.3% (10/12)	
specificity	100% (60/60)	100% (60/60)	
accuracy	97.2% (70/72)	97.2% (70/72)	*p* = 1.000
Pelvic lymph node metastasis (IIIC1)			
sensitivity	92.3% (24/26)	57.7% (15/26)	
specificity	97.8% (45/46)	97.8% (45/46)	
accuracy	95.8% (69/72)	83.3% (60/72)	*p* = 0.008 < 0.01
Paraaortic lymph node metastasis (IIIC2)			
sensitivity	100% (8/8)	75.0% (6/8)	
specificity	98.4% (63/64)	98.4% (63/64)	
accuracy	98.6% (71/72)	95.8% (69/72)	*p* = 0.480
Spread to adjacent organs such as the bladder or rectum (IVA)			
sensitivity	100% (4/4)	100% (4/4)	
specificity	100% (68/68)	100% (68/68)	
accuracy	100% (72/72)	100% (72/72)	*p* = 1
Spread to distant organs (IVB)			
sensitivity	100% (7/7)	71.4% (5/7)	
specificity	100% (65/65)	100% (65/65)	
accuracy	100% (72/72)	97.2% (70/72)	*p* = 0.480

**Table 3 diagnostics-11-00202-t003:** Comparison of ^18^F-FDG PET/MRI and CT for detecting lesion-based nodal metastasis.

	^18^F-FDG PET/MRI	CT	
sensitivity	83.3% (20/24)	29.2% (7/24)	*p* < 0.001
specificity	95.9% (256/267)	98.9% (264/267)	*p* < 0.05
accuracy	94.8% (276/291)	93.1% (271/291)	*p* = 0.424

## Data Availability

The datasets used and/or analyzed during the current study are available from the corresponding author on reasonable request.

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
