# Peer review of "Diagnostic Value of ^18^F-FDG PET/MRI for Revised 2018 FIGO Staging in Patients with Cervical Cancer"

_diagnostics, 2021, doi:10.3390/diagnostics11020202_

Round 1

Reviewer 1 Report

This is an interesting retrospective study by Tsuyoshi et al. on the role of FDG PET/MRI for the staging of cervical cancer patients. The main findings of the study are the following:

  • FDG PET/MRI showed a higher patient-based accuracy for metastasis to pelvic lymph nodes than CT
  • FDG PET/MRI showed a higher lesion-based sensitivity for lymph nodes than CT,
  • After excluding patients diagnosed by conization, accuracy for revised FIGO staging 2018 was significantly better for FDG PET/MRI than for CT and MRI.

Based on these findings, the authors conclude that PET/MRI may provide an optimal diagnostic strategy for preoperative staging of cervical cancer patients.

Despite its retrospective nature, the study design and methodology are robust. The results of the analysis are interesting. The manuscript is well-written and can easily be followed.

I believe that the manuscript could be accepted after the following minor points are addressed:

  • ln 113-120, MRI protocol: were any advanced MR techniques (e.g. DCE-MRI, DWI) applied?
  • ln 185-186. Why were these patients excluded?
  • ln 211: I believe the p value is wrong. It should be p=1. Correct the respective mistake also in Table 2 in pg12.
  • Figures: Due to the very similar signal of urine in the bladder and cervical cancer, I recommend the authors to use arrows also in the MIP images to delineate the tumor.
  • Minor spelling errors throughout the text.

Author Response

We wish to thank the reviewers for the time they have taken to assess the acceptability of our manuscript, and for the helpful suggestions provided. We considered their concerns as we revised our manuscript. We believe that the revised manuscript is greatly improved. Our responses to the specific comments from each reviewer are provided on the subsequent pages.

Reviewer reports:

Reviewer #1:

This is an interesting retrospective study by Tsuyoshi et al. on the role of FDG PET/MRI for the staging of cervical cancer patients. The main findings of the study are the following:

FDG PET/MRI showed a higher patient-based accuracy for metastasis to pelvic lymph nodes than CT

FDG PET/MRI showed a higher lesion-based sensitivity for lymph nodes than CT,

After excluding patients diagnosed by conization, accuracy for revised FIGO staging 2018 was significantly better for FDG PET/MRI than for CT and MRI.

Based on these findings, the authors conclude that PET/MRI may provide an optimal diagnostic strategy for preoperative staging of cervical cancer patients.

Despite its retrospective nature, the study design and methodology are robust. The results of the analysis are interesting. The manuscript is well-written and can easily be followed.

I believe that the manuscript could be accepted after the following minor points are addressed:

ln 113-120, MRI protocol: were any advanced MR techniques (e.g. DCE-MRI, DWI) applied?

Response: We appreciate this helpful suggestion. As advised, DCE-MRI or DWI have been reported to improve the diagnostic value in the local invasion or the lymph nodes metastasis.

(Woo S, et al. Magnetic resonance imaging for detection of parametrial invasion in cervical cancer: An updated systematic review and meta-analysis of the literature between 2012 and 2016. Eur Radiol. 2018 28:530-541.

Liu B, et al. A Comprehensive Comparison of CT, MRI, Positron Emission Tomography or Positron Emission Tomography/CT, and Diffusion Weighted Imaging-MRI for Detecting the Lymph Nodes Metastases in Patients with Cervical Cancer: A Meta-Analysis Based on 67 Studies. Gynecol Obstet Invest. 2017 82:209-222.

Akita A, et al. Comparison of T2-weighted and contrast-enhanced T1-weighted MR imaging at 1.5 T for assessing the local extent of cervical carcinoma. Eur Radiol 2011 21:1850-1857.)

However, no sufficient universal consensus has been reached in the use of these advanced MR techniques, whereas conventional MRI has been reported that it can accurately evaluate the distance between the tumor and the internal os, as well as stromal infiltration, and performs well in diagnosing the parametrial invasion.

(Xiao M, et al. Diagnostic performance of MR imaging in evaluating prognostic factors in patients with cervical cancer: a meta-analysis. Eur Radiol. 2020 30:1405-1418.)

Therefore, we focused on the T2WI in the present study.

ln 185-186. Why were these patients excluded?

Response: We appreciate this helpful suggestion. In the patients diagnosed by conization, cancer lesion is detected only by microscopy (FIGO IA) and cannot be detected by any imaging modalities. In the present study, we focused on the diagnostic value of 18F-FDG PET/MRI in the revised 2018 FIGO staging which recognizes metastatic or recurrence risk by including a greater number of tumor size subdivisions (IB1, IB2, and IB3) and by taking into account the status of regional lymph nodes detected radiographically or pathologically (IIIC1 and IIIC2). Therefore, we also showed the diagnostic value after excluding patients diagnosed by conization in addition to the diagnostic value for all patients.

According to the advice from the reviewer, we have revised our Results section. The following sentences “When we excluded patients diagnosed by conization to focus on the revision points, the accuracy of the revised 2018 FIGO staging was also significantly better for 18F-FDG PET/MRI (82.1%, 46/56) than for MRI/CT (60.7%, 34/56) (p <0.01).” have been included on line 2 of page 6 in the 3.3. Revised FIGO staging subsection of the Results section.

ln 211: I believe the p value is wrong. It should be p=1. Correct the respective mistake also in Table 2 in pg12.

Response: We appreciate this helpful suggestion. As suggested, we have revised the p value in the 3.3. Revised FIGO staging subsection of the Results section in page 6 and Table 2 in page 12.

According to the advice from the reviewer, we have revised our Results section. The following sentences “Sensitivity, specificity, and accuracy for growth into adjacent organs such as the bladder or rectum were 100%, 100%, and 100% for both 18F-FDG PET/MRI and MRI (p = 1).” have been included on line 29 of page 6 in the 3.3. Revised FIGO staging subsection of the Results section.

Figures: Due to the very similar signal of urine in the bladder and cervical cancer, I recommend the authors to use arrows also in the MIP images to delineate the tumor.

Response: We appreciate this helpful suggestion. As suggested, we have added the arrows to delineate the tumor in the MIP images in all Figures and revised the Figure legends.

According to the advice from the reviewer, we have revised our Figure legends. The following sentences “a. 18F-FDG PET image shows FDG uptake by tumor in the cervical cervix (arrow) and pelvic lymph nodes.” have been included on line 2 of page 7 in the Figure 1, and “a. 18F-FDG PET image shows FDG uptake by tumor in the cervical cervix (arrow) and a right inguinal lymph node.” have been included on line 2 of page 8 in the Figure 2, and “a. 18F-FDG PET image shows FDG uptake by tumor in the cervical cervix (arrow) and right pelvic and left inguinal lymph nodes.” have been included on line 2 of page 9 in the Figure 3, and “a. 18F-FDG PET image FDG uptake by tumor in the cervical cervix (arrow) and pelvic and paraaortic lymph nodes.” have been included on line 2 of page 10 in the Figure 4, and “a. 18F-FDG PET image shows FDG uptake by tumor in the cervical cervix (arrow) and a lung nodule.” have been included on line 2 of page 11 in the Figure 5.

Minor spelling errors throughout the text.

Response: We appreciate this helpful suggestion. As suggested, we have gone through the text and revised spelling errors.

Reviewer 2 Report

Hideaki Tsuyoshi et al. provided the manuscript entitled " Diagnostic value of 18F-FDG PET/MRI for revised 2018 FIGO  staging in patients with cervical cancer" The writing and presentation of this paper is well, and the topic would be important and interesting to whom in the related field. A minor request is that the abbreviation of some terms need to be described in full name, such as FIGO. The authors should consider to provide more contemporary references if available.

Author Response

We wish to thank the reviewers for the time they have taken to assess the acceptability of our manuscript, and for the helpful suggestions provided. We considered their concerns as we revised our manuscript. We believe that the revised manuscript is greatly improved. Our responses to the specific comments from each reviewer are provided on the subsequent pages.

Reviewer reports:

Reviewer #2:

Hideaki Tsuyoshi et al. provided the manuscript entitled " Diagnostic value of 18F-FDG PET/MRI for revised 2018 FIGO staging in patients with cervical cancer" The writing and presentation of this paper is well, and the topic would be important and interesting to whom in the related field. A minor request is that the abbreviation of some terms need to be described in full name, such as FIGO. The authors should consider to provide more contemporary references if available.

Response: We appreciate this helpful suggestion. As suggested, we have provided full name to the abbreviation of terms including FIGO, and 18F-FDG. Moreover, in the reference 1 and 10, we have changed the old references to the related contemporary ones.
